# A Novel AI Approach for Assessing Stress Levels in Patients with Type 2 Diabetes Mellitus Based on the Acquisition of Physiological Parameters Acquired during Daily Life

**DOI:** 10.3390/s24134175

**Published:** 2024-06-27

**Authors:** Gonçalo Ribeiro, João Monge, Octavian Postolache, José Miguel Dias Pereira

**Affiliations:** 1Department of Information Science and Technology, Iscte—Instituto Universitário de Lisboa, Av. das Forças Armadas, 1649-026 Lisbon, Portugal; goncalo_tome_ribeiro@iscte-iul.pt (G.R.); jpdme@iscte-iul.pt (J.M.); 2Instituto de Telecomunicações (IT), Instituto Superior Técnico, North Tower, 10th Floor, Av. Rovisco Pais 1, 1049-001 Lisbon, Portugal; 3Instituto de Telecomunicações, 3810-193 Aveiro, Portugal; dias.pereira@estsetubal.ips.pt; 4Instituto Politécnico de Setúbal, Escola Superior de Tecnologia de Setúbal, 2910-761 Setúbal, Portugal

**Keywords:** blood glucose monitoring, fuzzy logic, mobile application, photoplethysmography, physiological parameters extraction, stress assessment, wearable devices

## Abstract

Stress is the inherent sensation of being unable to handle demands and occurrences. If not properly managed, stress can develop into a chronic condition, leading to the onset of additional chronic health issues, such as cardiovascular illnesses and diabetes. Various stress meters have been suggested in the past, along with diverse approaches for its estimation. However, in the case of more serious health issues, such as hypertension and diabetes, the results can be significantly improved. This study presents the design and implementation of a distributed wearable-sensor computing platform with multiple channels. The platform aims to estimate the stress levels in diabetes patients by utilizing a fuzzy logic algorithm that is based on the assessment of several physiological indicators. Additionally, a mobile application was created to monitor the users’ stress levels and integrate data on their blood pressure and blood glucose levels. To obtain better performance metrics, validation experiments were carried out using a medical database containing data from 128 patients with chronic diabetes, and the initial results are presented in this study.

## 1. Introduction

Stress is a pervasive condition that affects an individual’s physical, mental, and emotional well-being. It has an impact on sleep patterns, dietary habits, self-perception, and cognitive functions. Individuals experiencing comorbid stress and mental disorder face difficulties in achieving prompt recovery. Without treatment, the symptoms can persist for extended periods of time, ranging from weeks to months or even years. Many people face challenging barriers in life that may clash or intersect, leading to stress that can have negative effects on their well-being. The workplace, interpersonal, and domestic factors can all contribute to the onset of stress.

Stress can manifest both physically and cognitively, resulting in alterations in multiple physiological reactions and potentially leading to pathological conditions. These can have a detrimental impact on an individual’s ability to focus and be productive, particularly by disrupting task execution. Over time, stress can exacerbate cognitive deficiencies and contribute to several issues, including diabetes, high blood pressure (BP), increased vulnerability to substance addiction, weakened immune system, and depression [1,2].

Effective stress management is crucial for sustaining optimal health. Currently, self-assessment questionnaires such as the Perceived Stress Scale (PSS) [3] are commonly used to measure stress levels. However, this method is not very practical. Technology can play a vital role in the evaluation of stress. Various suggestions have emerged, mostly utilizing smart bands, smartphones, and wearables, to acquire physiological measurements that are associated with stress detection, such as the heart rate (HR), heart rate variability (HRV), and galvanic skin response (GSR), among others. No comprehensive solution for acquiring several parameters has been suggested yet due to various challenges, such as limited battery life, interference from noise, device compatibility issues, system portability, user acceptance, and accuracy concerns.

This study presents significant advancements to the previous research conducted by the same authors [4]. The previous work involved the development of a multichannel sensory system that focused on acquiring and processing photoplethysmography (PPG) signals. The novelty of this study lies in the substantial expansion of the system’s capabilities by incorporating the acquisition of GSR and body temperature, two parameters closely associated with stress. Moreover, the innovation in this study includes the addition of new channels for monitoring BP and blood glucose levels, which enhances the system’s ability to cater to the needs of individuals with diabetes and hypertension. These novel channels significantly broaden the scope of the physiological data collected, making the system more comprehensive.

Furthermore, the authors have integrated sophisticated algorithms for the estimation of physiological parameters and applied machine learning techniques (fuzzy logic) for the classification of stress levels, demonstrating a leap towards intelligent health monitoring. The proposed advancements also include enhancements to both the intelligent stress assessment system and the developed mobile application. These innovations are designed to greatly assist patients in their daily lives, emphasizing practical application and user experience. Overall, this study not only introduces novel monitoring parameters but also leverages advanced analytical techniques to provide a more robust and user-friendly system for health monitoring.

The structure of this article is as follows. The history of health monitoring and stress assessment is discussed in Section 2, as well as the connections and effects between stress, blood pressure, and glucose levels. The methodology employed is described in Section 3, including the materials and tasks that were performed. The findings acquired through the executed experimental procedures are elaborated upon in Section 4. The conclusions and future work are presented in Section 5.

## 2. Related Work

Stress is a state of tension or pressure that results from everyday experiences or situations. It has a direct impact on the management of chronic diseases, such as heart disease, asthma, obesity, diabetes, headaches, depression and anxiety, gastrointestinal problems, Alzheimer’s disease, accelerated aging, premature death, among others [5]. Of these, the close link between hypertension and diabetes associated with stress stands out. Moreover, stress can be a significant burden on individuals’ lives. Therefore, it is crucial to effectively manage stress levels.

### 2.1. Impact of Stress on Diabetes

The main endocrine reactions to stress are glucocorticoids and catecholamines. Although these compounds do not have any negative effects in the short term, they might gradually alter the balance of glucose in the body by hindering the muscles and tissues from absorbing and using glucose. The prolonged elevation of blood sugar levels caused by this disruption in glucose regulation can lead to insulin resistance and the development of type 2 diabetes [1]. In addition, it can lead to a gradual decrease in muscle mass and the accumulation of visceral fat [6].

Metabolic abnormalities are recognized as products of chronic stress and obesity. Insulin resistance is the outcome of this metabolic anomaly. Survival under stress is facilitated by physiological responses that have been maintained through evolution, such as hyperglycemia and insulin resistance [7]. White and red blood cells, the reticuloendothelial system, the central and peripheral nervous systems, and the bone marrow are the largest consumers of glucose without requiring insulin [8]. The immune system and the brain are also fueled by stress hyperglycemia in response to injury, infection, and stress [9,10]. Acute hyperglycemia during stress may be advantageous to the body and an integral part of its evolutionary process, as discussed previously. However, chronic stress induces insulin resistance because of various factors, including chronic hyperglycemia. Thus, persistent stress causes type 2 diabetes.

Type 2 diabetes is defined by the presence of an excessive amount of glucose in the bloodstream, which cannot be effectively used by the body due to a condition called insulin resistance. It can lead to significant challenges, disturbance of daily living, and persistent health problems. Possible effects include neuropathy, nephropathy, cataracts, glaucoma, increased susceptibility to severe fungal infections, delayed wound healing, auditory impairment, and other similar issues. Patients should be motivated to participate in daily activities and adhere to regular follow-up consultations, as prevention is always better, especially considering that the illness is predominantly associated with obesity and can be reversed with dietary changes, physical exercise, and medications [11].

### 2.2. Stress and Blood Pressure

Concerning the relationship between stress and BP, it is important to understand that the human body produces an abundance of hormones because of stress. These hormones transiently elevate blood pressure, resulting in increased HR and the constriction of blood vessels. Fluctuations in the size of blood vessels might increase the likelihood of experiencing health issues, such as hypertension (which is characterized by high blood pressure caused by narrowed blood vessels) and hypotension (which is associated with low blood pressure due to widened blood vessels) [12]. Clinical trials have examined the impact of cognitive behavioral approaches on blood pressure [13].

### 2.3. Correlation between Diabetes and Blood Pressure

Diabetes is a long-term condition caused by insufficient insulin, which can be due to a lack of insulin production or insensitivity to insulin [14]. It is linked to various complications, primarily caused by consistently high blood glucose levels, which can result in the development of microvascular diseases (damage to small blood vessels) and macrovascular diseases (damage to arteries). Diabetic complications, such as neuropathy (nerve damage), retinopathy (eye disease), and nephropathy (kidney disease), occur due to the vascular damage that affects organs and tissues [15].

Previous studies have confirmed a connection between hypertension and diabetes [16]. This connection is not only because individuals with diabetes are more likely to develop hypertension compared to those without diabetes [17], but also because it increases the likelihood of these two illnesses occurring together. Therefore, the simultaneous presence of these two circumstances may result in the emergence of novel public health concerns, such as an elevated vulnerability to stroke [18], a notable rise in disruptions to the body’s natural sleep–wake cycle, and harm to specific organs [19]. Furthermore, in patients with both diabetes and hypertension, insulin resistance is a separate risk factor for ischemic cerebral infarction, particularly lacunar infarction [20].

### 2.4. Stress Management

Managing stress requires maintaining an active, wholesome, and enjoyable lifestyle. Lifestyle modifications, exercise, meditation, yoga, relaxation, and the modification of behavior and attitude are all components of stress treatment programs. Changes in perspective are vital for stress management. This necessitates an optimistic mindset and a positive perspective. Rather than a ritual, stress management is a holistic process that balances physical, psychological, emotional, and spiritual health. It enhances concentration, serenity, productivity, and perseverance.

Continuous monitoring of stress levels is essential for effective stress management. The golden standard for assessing stress is self-assessment questionnaires, but new methods have been proposed using non-invasive physiological assessment methods, including electrocardiography (ECG), HR, GSR, and BP, as well as respiratory activity [21,22,23].

Additionally, stress monitoring using face recognition technology has been documented [24,25]. This approach leverages the functionalities of mobile devices, including tablets and smartphones [26,27]. An illustration of this concept is face recognition, which attempts to determine an individual’s affective state through the comparison of a captured facial expression and a database comprising instances of facial expressions that correspond to a predetermined meaning.

## 3. Materials and Methods

This section provides an overview of the healthcare monitoring system that has been proposed, along with the experimental procedure used to validate the system and the fuzzy logic model that has been developed for stress assessment. Considerations are also given to the classification of blood glucose levels.

### 3.1. Proposed Healthcare System

The physiological parameters gathered by the proposed system can be used to evaluate the stress level and overall health of the user. The system architecture is illustrated in Figure 1. To provide greater specificity, the architecture of the system comprises the subsequent layers:Sensing system: wearable device developed for the acquisition of physiological parameters and the establishment of a communication channel with the mobile application, allowing users to be authenticated and data to be stored locally or remotely, depending on the availability of the network.Mobile application: A user interface is provided to access the data stored in the database. This interface allows the entry of data such as glucose and blood pressure levels. It also includes a feature to classify stress levels, user authentication, and clinical advice based on the analysis of physiological indicators.Database: tasked with the remote storage of user-specific data and information. The Google NoSQL database “Firebase” was selected for the proposed system because of its benefits in developing mobile and web applications, including its interoperability with IOS, Android, Web, Unity, and C++.Glucose monitor: Element NEO is a straightforward glucose monitor designed to assist users in managing diabetes by incorporating functions that reduce the likelihood of complications associated with this condition. With its backlit display, ergonomic design, and illumination at the test strip insertion point, this device is suitable for all types of users. This device has been certified by Common Era (CE) [28].Blood pressure monitor: measures BP and pulse rate and allows users to check the results directly on the screen, which changes color depending on the level of blood pressure (red, yellow, and green). This device has been certified by the U.S. Food and Drug Administration (FDA) and CE [29].

#### 3.1.1. Sensing System

The proposed sensing system builds upon the prior research [4] and utilizes PPG signal processing to estimate HR, pulse rate variability (PRV), blood oxygen saturation (SpO2), and respiratory rate (RR). This system also has channels for gathering physiological indicators that are essential to monitoring health status and classifying stress levels, such as GSR and body temperature (BT). The architecture of the sensing system is depicted in Figure 2. The system’s sampling rate is 300 samples per second, enabling it to accurately reproduce the PPG signal. To determine the values of HR, PRV, RR, and SpO2, it is crucial to identify the highest points in the signals. Therefore, the system collects two measurements of each physiological parameter every second.

The sensing system consists of an ESP32 microcontroller with a Dual-Core 32-bit CPU, a maximum clock speed of 240 MHz, a ROM memory capacity of 448 Kbytes, a RAM capacity of 520 Kbytes, and a flash memory capacity of 4 MB. Utilizing Digital Signal Processing (DSP), the PPG signal was analyzed to determine the highest, lowest, and average values. Additionally, the HR, PRV, RR, and SpO2 were estimated. The data were thereafter transmitted via Bluetooth 5.1 to the mobile application for data storage purposes.

The system incorporates a MAX30102 PPG sensor that enables the capture of the infrared component of the PPG signal, which is essential for estimating HR, PRV, and RR. Additionally, it captures the red-light component of the PPG signal, which is important for estimating SpO2. Additionally, a GSR sensor was incorporated to measure changes in sweat gland activity caused by stress. The GY906 sensor accurately monitors infrared body temperature within a temperature range of −70 °C to +380 °C, with a precision of 0.5 °C, regardless of whether there is physical contact or not. The system is equipped with a real-time data display screen and Li-Po batteries, each offering up to 4 h of uninterrupted monitoring autonomy. The system components establish communication with the ESP32 through the Inter-Integrated Circuit (I2C) protocol.

#### 3.1.2. Database

As technology advances, intelligent systems are more dependent on efficient and automated data processing methods, such as machine learning (ML) approaches, neural networks, deep learning, data mining, and others. For these technologies to carry out their functions effectively and precisely, they require substantial quantities of data kept within databases. Therefore, selecting the most appropriate database for a system is of utmost importance. Some databases are specifically designed for processing and implementing algorithms, while others are more advantageous for quick access and customized organization based on data type [30].

The Google “Firebase” platform was selected for implementing this system because of its dynamic database capabilities, which offer significant benefits for mobile and web application development. It is compatible with IOS, Android, Web, Unity, and C++. This platform offers both real-time access and a high level of interaction with cloud storage, use of ML techniques, rapid and secure authentication methods, and other features [31].

Figure 3 demonstrates how Firebase is configured to store data. Each user is assigned a unique identification (ID), and the data transmitted from the detection system and both monitors (glucose and blood pressure) are accompanied by a timestamp. Firebase organizes the data within the nodes associated with each ID according to real-time data and the data stored previously.

#### 3.1.3. Mobile Application

The user interface is built upon an Android mobile application, as depicted in Figure 4. The mobile application enables user authentication, continuous monitoring of their physiological parameters, input of glucose and blood pressure readings, and the classification of stress into three categories using the fuzzy logic model outlined in Section 3.2, “Stress Assessment”.

As shown in Figure 4a, the application displays this classification and physiological parameter values. Interactive blue and red spheres provide physiological parameter information. In Figure 4d, each physiological parameter’s real-time progression is graphically illustrated. In Figure 4b, users can access their daily and monthly physiological parameter averages. As shown in Figure 4c, these averages can be displayed on graphs with colors according to physiological parameter classifications.

#### 3.1.4. Glucose and Blood Pressure Monitors

The glucose monitor Element NEO is a device designed for the self-monitoring of blood glucose levels, commonly used by individuals with diabetes. It is compact and portable, making it easy to carry, and features a clear, easy-to-read LCD screen that displays blood glucose readings and other relevant information.

This monitor provides quick and accurate blood glucose readings, often within seconds, and can store a significant number of past readings, allowing users to track their blood glucose levels over time. It uses specific test strips that require a small blood sample for measurement and may support testing on alternative sites such as the forearm or palm, which can be less painful than fingertip testing. The automatic coding feature eliminates the need for the manual coding of test strips, reducing user error.

Element NEO is designed for easy operation, with a simple interface and a minimal number of buttons. It offers connectivity options such as USB or Bluetooth to transfer data to a computer or smartphone for more detailed analysis. Additionally, it can include features such as alarms to remind users to test their blood sugar at regular intervals.

This monitor is designed to provide highly accurate readings, which is crucial for effective diabetes management. Some models require calibration with control solutions to ensure accuracy, while others may be factory calibrated. It is typically powered by replaceable batteries, with a long battery life due to low power consumption.

Some models might have the capability to warn users if blood ketone levels are too high. Temperature compensation ensures accurate readings, even in varying temperature conditions, and hemoglobin correction allows some advanced models to adjust readings based on the hemoglobin levels in the blood.

Regarding the iHealth Track Blood Pressure Monitor, it is a device designed for measuring and tracking blood pressure at home. It has a sleek and compact design, making it easy to use and portable. The monitor features a large, backlit LCD screen that displays readings clearly, even in low light conditions.

This blood pressure monitor has a user-friendly interface with simple, one-button operation, making it easy to use for people of all ages. It can store multiple readings, allowing for individual tracking and comparison over time. The device comes with a comfortable, adjustable cuff that fits a wide range of arm sizes. It also offers Bluetooth connectivity, allowing it to sync with the iHealth MyVitals app, enabling users to store and track their readings on their smartphones.

The iHealth Track provides accurate systolic and diastolic blood pressure readings, as well as pulse rate. It includes color-coded indicators to quickly identify normal, prehypertension, and hypertension levels based on World Health Organization (WHO) standards. The device can also detect irregular heartbeats and alert the user if an irregular pattern is found during measurement.

### 3.2. Stress Assessment

The proposed algorithm for stress assessment relies on collecting stress-related physiological parameters, including HR, PRV, RR, SpO2, GSR, and BT, as described in [4]. Additionally, BP has been incorporated into the model. The stress levels were classified using fuzzy logic, where we defined the reference values for each physiological parameter (HR [32], PRV [33], RR [34], SpO2 [35], GSR [36], BT [37], and BP [38]). The classification and type of membership function were determined based on Table 1, which displays the mean values that are representative of the overall healthy population, and they are equally applicable to individuals with chronic conditions. Variations in these parameters, for example for high-performance athletes, have not been considered due to the wide disparity in values. The aim of this work is to offer improvements in health care for the chronically ill, as well as stress management for the general population.

The membership functions used in the proposed model are categorized as the trapezoidal function Type R for “Low”, trapezoidal function Type L for “High”, and triangular function for “Normal”, as documented in [4], based on the physiological parameter’s configuration and classification, presented in Table 1.

To implement the fuzzy logic model designed for stress level classification, three rules were established in accordance with the examined correlation between physiological parameters and stress [21]. The rules are outlined in Table 2. Following this, the stress level is quantified using Equation (1). Note that in fuzzy logic, the final process of defuzzification requires the application of the maximum method or the centroid method, the latter being used. The variables *R*1, *R*2 and *R*3 refer to the indices obtained by applying the rules shown in Table 2, where each parameter of these rules refers to the membership degree, obtained through the membership functions identified previously and detailed in [4].
(1)$$\mathrm{Stress}=\frac{\left(R1\times 1\right)+\left(R2\times 2\right)+\left(R3\times 3\right)}{R1+R2+R3}$$

### 3.3. Blood Glucose Levels

Blood glucose levels are measured in milligrams per deciliter (mg/dL) and are a crucial category for carbohydrates in biology, as they play a vital function as a metabolic energy source for cells. Prolonged elevation of glucose levels typically indicates the presence of diabetes. Diabetes is a medical condition characterized by insufficient production of insulin by the body (type 1 diabetes) or inadequate response of the body’s cells to insulin (type 2 diabetes).

To diagnose diabetes, individuals usually undergo blood glucose and hemoglobin tests. However, the results of these tests may vary depending on whether the patient has fasted or eaten food during the day. Therefore, the fluctuations in blood glucose levels [39] are outlined in Table 3.

### 3.4. Experimental Procedure

In order to obtain better performance metrics, validation experiments were carried out using a public database [40] containing data from 128 patients with chronic diabetes. The dataset was collected from a variety of sources, including medical records, surveys, and interviews. The data were cleaned and processed to ensure their accuracy and integrity and are licensed under CC BY 4.0. The volunteers’ specific biometric information is shown below, in Table 4. All participants gave their verbal consent.

The primary aim of this study is to examine the impact of stress on individuals with chronic conditions, particularly those diagnosed with diabetes. Considering the correlation between diabetes and hypertension, a system that was previously created and verified in [4] has been modified to include two more channels. One channel is dedicated to monitoring blood glucose levels, while the other is focused on tracking blood pressure. Furthermore, the stress level assessment model used and validated in [4] was modified to incorporate a three-level scale and membership functions better suited to the characteristics of the data.

The experimental protocol established the simultaneous acquisition of stress, BP, and glucose data, which enabled the correlation between these three indicators. All data collection was conducted during a fasting state, taking into consideration that this is the optimal time of day for individuals with diabetes to obtain their glucose levels, unaffected by the impact of food intake.

To enhance this study, an additional phase was implemented for some participants, during which physiological parameters were collected four times each day, over a period of five consecutive days. The selection of these four time periods throughout the day was made in order to evaluate the impact of both meals and medicine intake. Consequently, during the first and second stages, all the physiological measurements were obtained following food intake and after the medication began to produce its effects, respectively. This enabled the evaluation of the effectiveness of the medication. In the third and fourth stages, we collected the physiological parameters both before and after food intake, respectively. This enabled us to evaluate the impact of food intake on individuals with diabetes.

## 4. Results and Discussion

This section provides an analysis of the findings pertaining to the correlation between blood glucose levels and stress levels, as well as blood glucose levels and BP values. Furthermore, a comparison is conducted on the reduction of the stress assessment model’s scale from five, as it was originally proposed in prior research [4], to three.

### 4.1. Stress Level Classification Model Update

In a previous study, a stress assessment model comprising five levels was utilized [4]. Nevertheless, it is important to mention that the extreme levels only consisted of residual data, which did not have any noticeable effect on the system’s performance, but did result in a small divergence in the accuracy of the model. Therefore, this study aimed to update these levels and thereby reduce them from five to three.

The performance of the model was evaluated using a multi-class confusion matrix, which is depicted in Figure 5. Note that the “True Classification” refers to the volunteers’ classification of their stress levels through a self-assessment questionnaire, while the “Predicted Classification” refers to the stress levels estimated by the proposed fuzzy logic algorithm. This enables the identification of true positives (TPs), true negatives (TNs), false positives (FPs), and false negatives (FNs). Furthermore, several metrics were used to analyze the performance of the proposed stress assessment model, as outlined in Table 5. In this case, sensitivity measures the proportion of TPs that are correctly identified by the model; specificity measures the proportion of TNs that are correctly identified by the model; precision measures the proportion of predicted positives that are correct; accuracy measures the overall correctness of the model; the F1 score is the harmonic mean of precision and sensitivity, providing a balance between the two.

Table 5 shows that updating the proposed fuzzy logic methodology produces satisfactory results in terms of precision, accuracy, sensitivity, and specificity, compared to the work carried out in [4], thus demonstrating that reducing the number of stress level classifications has improved the data, reducing dispersion, without this having influenced the system’s performance. Another motivating factor is comparing the accuracy of the proposed method with others, proposed in the more detailed literature review presented in [4]. To enhance the suggested model, it is necessary to carry out supplementary stress induction studies or explore alternative methods of inducing stress, such as noise or light.

### 4.2. Correlation between Stress and Glucose Levels

Regarding the correlation of stress with blood glucose levels, the results obtained were promising, allowing to identify a clear relationship between the two parameters. In Figure 6 below, it is possible to observe the evolution of the stress levels (classified from 1 to 3, as stipulated in the proposed model) and the blood glucose levels. The samples identify each of the 128 participants, clearly showing that when one increases, the other follows, and vice versa.

Considering a more specific case, in Figure 7 below, we have the acquisition of the physiological parameters for one of the participants who took part in the extra stage of the experimental protocol, namely the acquisition during four phases of the day over five consecutive days. Note that the timestamp refers to the day and time of acquisition.

The data acquired and presented in Figure 7 show that after food intake (the third and fourth phases), more precisely about forty minutes after eating, the human body begins the process of food digestion, which results in an increase in glucose levels. Note that this reaction is normal for anyone; however, for diabetics, this reaction can be quite accentuated, which is why they must take medication after the meal to counteract this effect. On the other hand, it takes about one hour for the medication to take effect (the first and second phases), leading to a reduction in glucose levels, suggesting that the medication is appropriately adjusted. Concerning the correlation between glucose and stress levels, it is clear that the increase in one is linked to the increase in the other, and vice versa.

### 4.3. Correlation between Glucose Levels and Blood Pressure

The correlation between blood glucose levels and blood pressure was also found to be promising, as the results enabled the identification of a distinct relationship between the two parameters, considering both the systolic and diastolic blood pressure components. In Figure 8 below, it is possible to observe the evolution of glucose levels and blood pressure, where the samples identify each of the 128 participants, showing that when one parameter increases, the same change is visible in the other parameter, and vice versa.

Considering a more specific case, in Figure 9 below we have the acquisition of the physiological parameters for one of the participants who took part in the extra stage of the experimental protocol, namely the acquisition during four phases of the day over five consecutive days. The timestamp refers to the day and time of acquisition.

As stated before, and shown in Figure 9, approximately forty minutes after food intake (between the third and fourth phases), the body begins the process of digesting the food, which results in an increase in glucose levels. However, it takes around one hour for the medicine to take effect (between the first and second phases), resulting in a decrease in glucose levels, which demonstrates that it is correctly adjusted. Concerning the correlation between glucose levels and blood pressure, it is clear that when one increases, the same is visible in the other parameter, and vice versa.

### 4.4. Correlation between Stress and Blood Pressure

In relation to the correlation of stress with blood pressure, the results obtained were promising, allowing to identify a clear relationship between the two parameters. In Figure 10 below, it is possible to observe the evolution of the stress levels (classified from 1 to 3, as stipulated in the proposed model) and the blood pressure represented here through its two components, i.e., systolic blood pressure and diastolic arterial pressure. Furthermore, the samples identify each of the 128 participants.

Considering a more specific case, in Figure 11 below, we have the acquisition of the physiological parameters for one of the participants who took part in the extra stage of the experimental protocol, namely the acquisition during four phases of the day over five consecutive days. Again, the timestamp refers to the day and time of acquisition.

Figure 11 shows that the highest blood pressure value of the day is also associated with the highest level of stress, thus corroborating the relationship between stress and blood pressure. It should be noted that eating can lead to changes in various physical parameters, which can indirectly influence stress levels; for example, after eating, blood flow directed to the digestive system can cause drowsiness or lethargy, which can interfere with productivity and cause stress. Eating also increases metabolic activity, temporarily increasing the heart rate and blood pressure, which can be stressful, especially for people with cardiovascular problems.

## 5. Conclusions

The multichannel sensing system that has been implemented offers an important level of mobility and interactivity, enabling the real-time visualization of health status parameters, such as stress levels. In pursuit of this objective, a mobile application was effectively designed and verified. Given that many patients with diabetes believe that controlling their well-being entails merely altering their daily routines, the way this work integrates a mobile application’s user-friendly interface and an efficient system with a significant number of crucial components and mechanisms for acquiring physiological parameters is an asset. This integration assists these patients to gain a better understanding of how their daily activities impact their health.

Significant conclusions were drawn from the experimental validation conducted to investigate the correlation between stress, diabetes, and blood pressure. These conclusions establish the relationship between these parameters and emphasize once more the importance of patients with diabetes incorporating the management of nervous system responses into their daily routines. Despite adhering to a healthy lifestyle and carefully monitoring their blood glucose levels, patients with diabetes should be more aware of the impact of stress.

Regarding future work, one of the next steps is to enhance the system’s robustness by perhaps incorporating new mechanisms and implementing changes. One of the goals is to replace the fuzzy logic technique with more powerful machine learning techniques. Furthermore, given that the purpose of this project is to assist individuals with diabetes in their everyday activities, it would be advantageous to prioritize the development of an intelligent and mobile interface that allows for seamless user interaction. In the future, it would be beneficial to incorporate features for measuring glucose levels and blood pressure directly into the system, rather than relying on the external devices currently available in the market.

## Figures and Tables

**Figure 1 sensors-24-04175-f001:**
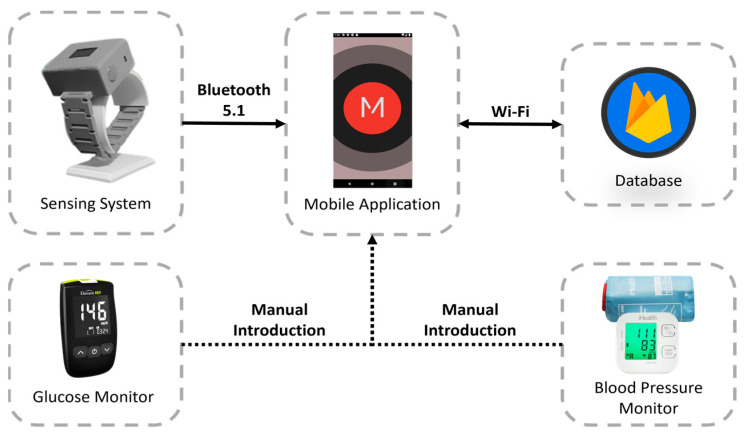
System architecture.

**Figure 2 sensors-24-04175-f002:**
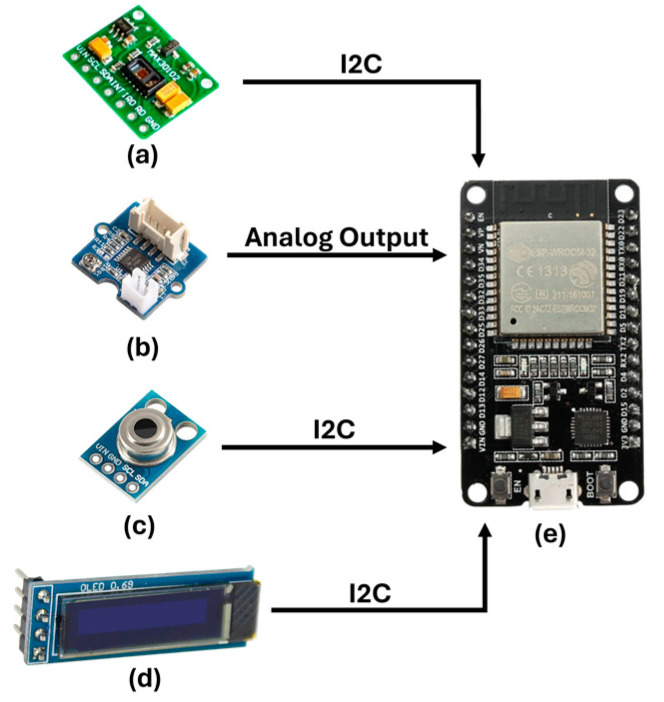
Sensing system architecture. (**a**) Photoplethysmography sensor. (**b**) Galvanic skin response sensor. (**c**) GY906 infrared temperature sensor. (**d**) Display. (**e**) ESP32 microcontroller.

**Figure 3 sensors-24-04175-f003:**
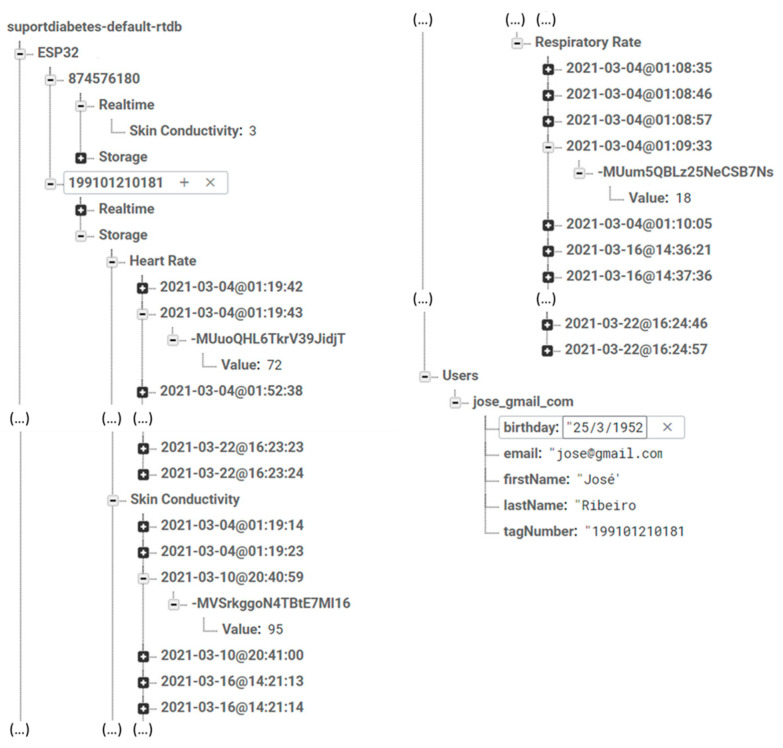
Firebase data storage architecture and exemplification. In the database structure, data are stored inside the “ESP32” node, while the “Users” node is used to store users’ personal information, allowing data to be linked to the respective users through “tagNumber”.

**Figure 4 sensors-24-04175-f004:**
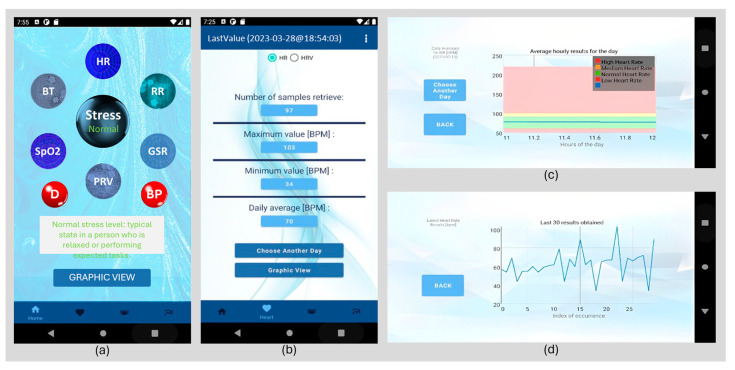
Main application screens. (**a**) Stress assessment and physiological parameters monitoring. (**b**) Physiological parameter averages by day and month. (**c**) Alternative representations of daily and monthly averages. (**d**) Physiological parameter real-time monitoring displayed on graphs.

**Figure 5 sensors-24-04175-f005:**
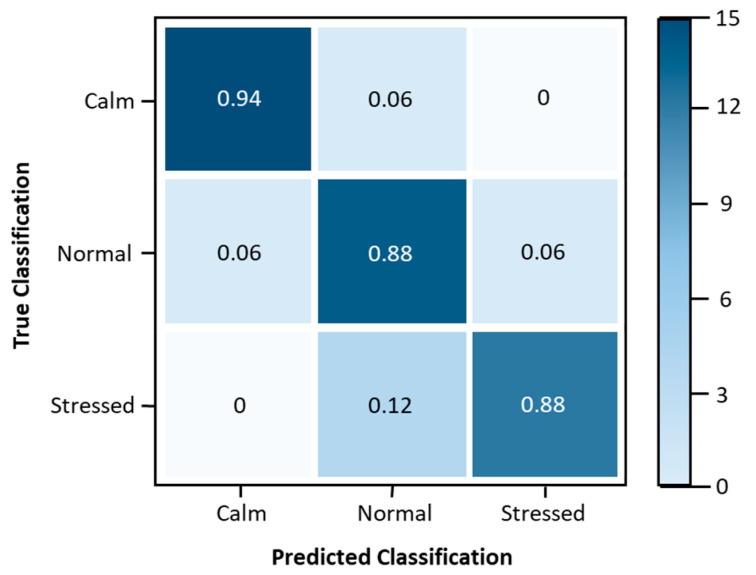
Confusion matrix with multiple classes for the stress classification model.

**Figure 6 sensors-24-04175-f006:**
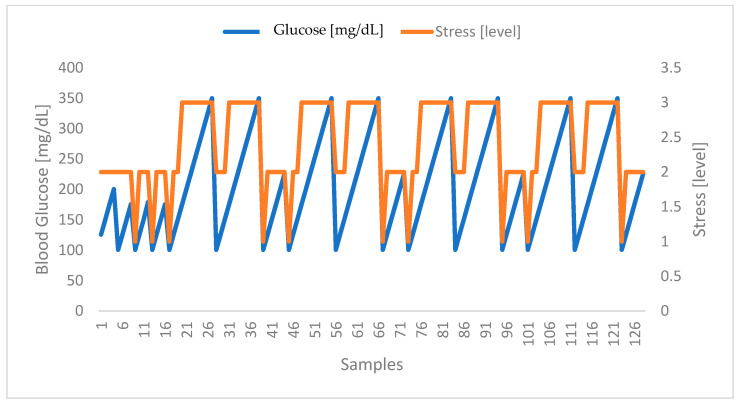
Correlation between stress and glucose levels. Highlighted in blue are the blood glucose values of the volunteers. In turn, highlighted in orange are the stress levels obtained at the same moments.

**Figure 7 sensors-24-04175-f007:**
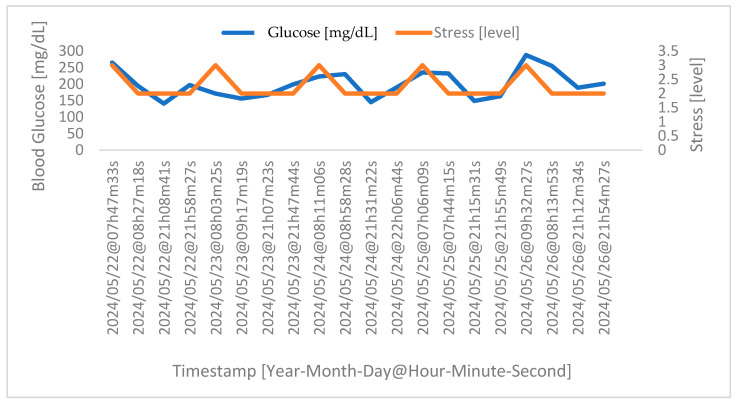
Correlation between stress and glucose levels for a random volunteer. Highlighted in blue are the blood glucose values of the volunteer. In turn, highlighted in orange are the stress levels.

**Figure 8 sensors-24-04175-f008:**
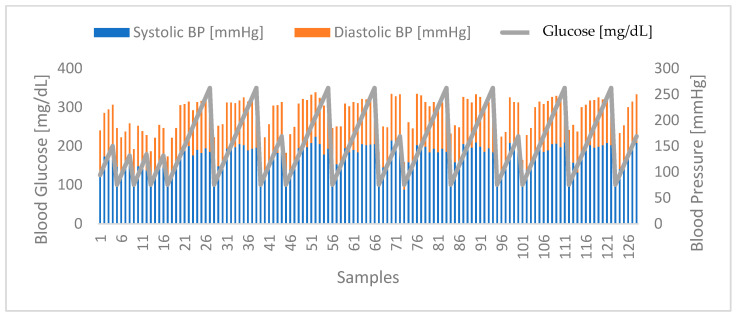
Correlation between glucose levels and blood pressure. Highlighted in blue are the systolic blood pressure values of the volunteers. In turn, highlighted in orange are the diastolic blood pressure values obtained at the same moments. Highlighted in grey are the glucose levels of the volunteers.

**Figure 9 sensors-24-04175-f009:**
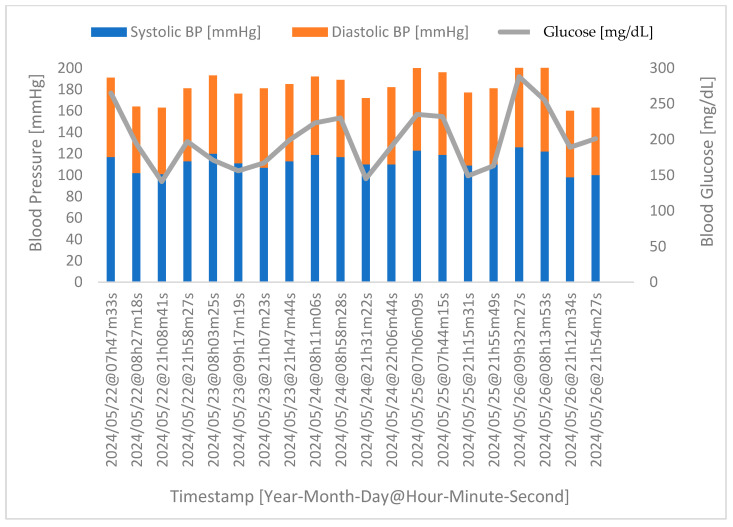
Correlation between glucose levels and blood pressure for a random volunteer. Highlighted in blue are the systolic blood pressure values of the volunteer. In turn, highlighted in orange are the diastolic blood pressure values. Highlighted in grey are the glucose levels.

**Figure 10 sensors-24-04175-f010:**
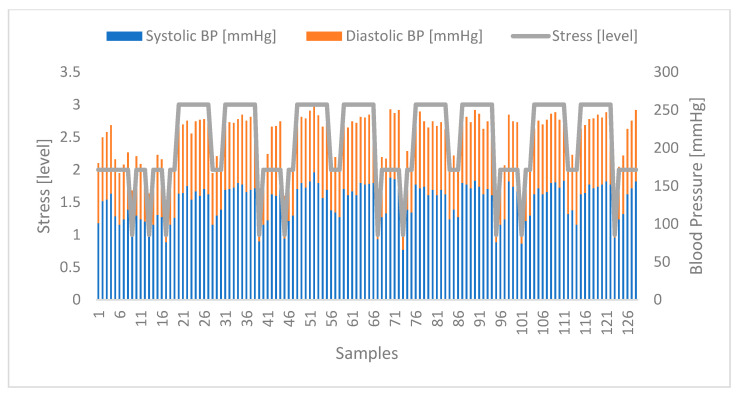
Correlation between stress and blood pressure. Highlighted in blue are the systolic blood pressure values of the volunteers. In turn, highlighted in orange are the diastolic blood pressure values obtained in the same moments. Highlighted in grey are the stress levels of the volunteers.

**Figure 11 sensors-24-04175-f011:**
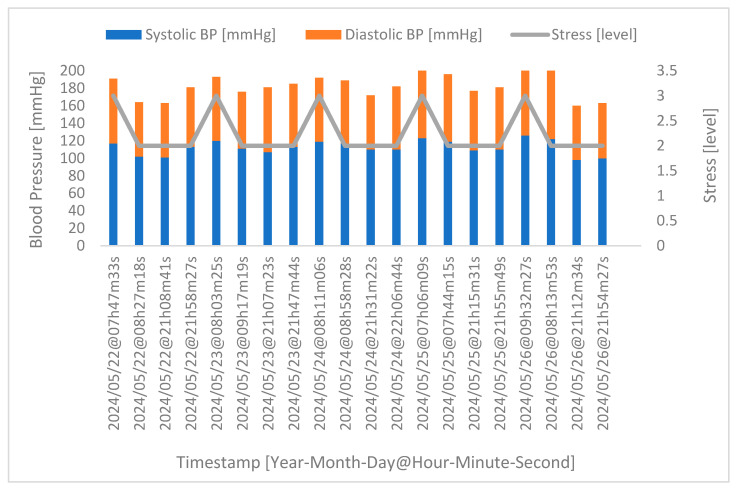
Correlation between stress and blood pressure for a random volunteer. Highlighted in blue are the systolic blood pressure values of the volunteer. In turn, highlighted in orange are the diastolic blood pressure values. Highlighted in grey are the stress levels.

**Table 1 sensors-24-04175-t001:** Physiological parameters treatment.

Parameter	Classification Based on Parameter Reference Values		
	Low	Normal	High
HR [beat-per-minute]	<60	60–90	>90
PRV ^1^ [ms]	<32	32–77	>77
RR [breath-per-minute]	<12	12–18	>18
SpO2 [%]	<97	97–99	>99
GSR [kOhm]	<30	30–50	>50
BT [°C]	<36.5	36.0–37.5	>37.5
Systolic BP [mmHg]	<90	90–120	>120
Diastolic BP [mmHg]	<60	60–80	>80

^1^ The PRV reference values are determined using the root mean square of successive differences (RMSSD) and include the whole range of possible values for all healthy age groups, apart from children.

**Table 2 sensors-24-04175-t002:** Fuzzy logic model for stress assessment.

Status	Rules (R)
Calm	*R*1 = Low(HR) Λ Low(HRV) Λ Low(RR) Λ High(SpO2) Λ High(GSR) Λ High(BT) Λ Low(Systolic BP) Λ Low(Diastolic BP)
Normal	*R*2 = Normal(HR) Λ Normal(HRV) Λ Normal(RR) Λ Normal(SpO2) Λ Normal(GSR) Λ Normal(BT) Λ Normal(Systolic BP) Λ Normal(Diastolic BP)
Stressed	*R*3 = High(HR) Λ High(HRV) Λ High(RR) Λ Low(SpO2) Λ Low(GSR) Λ Low(BT) Λ High(Systolic BP) Λ High(Diastolic BP)

**Table 3 sensors-24-04175-t003:** Tabulated values for blood glucose levels.

Glucose Levels in Fasting [mg/dL]	Glucose Levels after Meals [mg/dL]	Blood Glucose Category
<70	<70	Low blood glucose levels associated with hypoglycemia.
70–99	70–139	Normal blood glucose levels.
100–125	140–199	Pre-diabetes status.
>125	>199	Diabetes diagnosis.

**Table 4 sensors-24-04175-t004:** Volunteers’ specific biometric information.

	Male Gender	Female Gender	Total
Volunteers	68	60	128
Age Range	12–75	17–75	12–75
Average Age	41	43	42
Standard Deviation of Ages	17	16	17

**Table 5 sensors-24-04175-t005:** The application of performance metrics to the stress model.

Metric	Stress Levels Classification		
	Calm	Normal	Stressed
Sensitivity	0.94	0.82	0.93
Specificity	0.97	0.94	0.94
Precision	0.94	0.88	0.88
Accuracy	0.96	0.90	0.94
F1 Score	0.94	0.85	0.90

## Data Availability

Data are contained within the article.

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
