# Peer review of "A Novel AI Approach for Assessing Stress Levels in Patients with Type 2 Diabetes Mellitus Based on the Acquisition of Physiological Parameters Acquired during Daily Life"

_sensors, 2024, doi:10.3390/s24134175_

Round 1

Reviewer 1 Report

Comments and Suggestions for Authors

In this paper, it investigates the design and implementation of a distributed wearable sensor computing platform that utilizes fuzzy logic algorithms to assess stress levels in patients with type 2 diabetes. However, there are still some problems in the article that need to be corrected. It is recommended to accept this paper after revision.

1.     The background for the fuzzy logic algorithm should be fully provided and discussed, and the novelty should be highlighted in more detail.

2.     The " 3.1. Proposed Healthcare System" section on page 4 should adequately provide a description of the overall system.

3.     Further clarification should be provided under Formula 1 on page 7, and under Table 2.

4.     In the section " 4.1. Stress Level Classification Model Update", you refer to "satisfactory results". Objects of comparison should be provided and discussed adequately.

5.     In section 3.1.2, the paper mentions "fast and secure authentication methods using ML technology" and also mentions "a new approach to artificial intelligence" in the title. Please fully provide and discuss the application of the system in ML.

7.     The picture in the text is fuzzy, such as Figure 4, please check it carefully.

Comments on the Quality of English Language

There are many typographical errors. For example, on page 9, should Figure 4 be Figure 5? Please check the chart numbers and manuscript carefully.

Author Response

“(1) Please kindly confirm that the Article Processing Charges (0 CHF) will apply to your article, if accepted for publication.”

We greatly appreciate your concern. This issue has already been addressed and was identified via the link provided in the corresponding email.

“(2) We noticed that the corresponding author in the submission system (Prof. Octavian Postolache and Prof. José Dias Pereira) is different from the manuscript (Prof. Octavian Postolache), please confirm with us which one is correct.”

We appreciate your comment. This issue has already been addressed and was identified via the link provided in the corresponding email.

“(3) As your research involves diabetes patients, approval from an ethics committee should be obtained. Please get the permission and send a scanned copy to us.”

We appreciate the concern and the reason for your request. The approach presented in this work was specifically developed to gather data from individuals suffering from chronic diabetes. However, due to the limited number of eligible participants for this study, it became necessary to utilise a publicly available database that contained information from 128 patients with chronic diabetes. The dataset was obtained from many sources, such as medical records, surveys, and interviews, and is licenced under CC BY 4.0. This licence provides valuable information regarding the reliability of the data, particularly in terms of ethical approval. In this specific instance, the data provided by medical institutions, which is both licenced and publicly available, ensures the ethical approval that is being requested with this matter. Nevertheless, the manuscript lacked clarity regarding this information, prompting the need for adjustments to enhance the explicitness of the data's provenance. As such, the following information has been added to the Abstract:

“To obtain better performance metrics, validation experiments were carried out using a medical database containing data from 128 patients with chronic diabetes, and the initial results are presented in the study.”

In addition, the following information has been added in Subsection 3.4 Experimental Procedure, also referring to the public database in question, to which the reference has also been added in the Reference Section under the number 39:

“In order to obtain better performance metrics, validation experiments were carried out using a public database [39] containing data from 128 patients with chronic diabetes. The dataset was collected from a variety of sources, including medical records, surveys, and interviews. The data was cleaned and processed to ensure its accuracy and integrity and is licensed under CC BY 4.0.”

We hope that this has successfully addressed the concern pointed out by the Editor.

“The background for the fuzzy logic algorithm should be fully provided and discussed, and the novelty should be highlighted in more detail.”

We greatly appreciate your comment. With regard to the background of the Fuzzy Logic algorithm, as mentioned in the text, this is detailed in the previous work and referenced as [4]. The purpose is not to repeat the basic concepts that have already been discussed in the mentioned work, thus avoiding redundancy. One of the differences between the previous work and this one has to do with updating the classification criteria and not the basis of the algorithm. Consequently, it was deemed unnecessary to incorporate this background information, but instead, reference the documentation in [4]. Regarding the need to highlight in more detail the novelty, new information has been added in Section 1. Introduction, as follows:

“This study presents significant advancements to the previous research conducted by the same authors [4]. The previous work involved the development of a multichannel sensory system that focused on acquiring and processing Photoplethysmography (PPG) signals. The novelty of this study lies in the substantial expansion of the system's capabilities by incorporating the acquisition of GSR and body temperature, two parameters closely associated with stress. Moreover, the innovation in this study includes the addition of new channels for monitoring BP and blood glucose levels, which enhances the system's ability to cater to the needs of individuals with diabetes and hypertension. These novel channels significantly broaden the scope of physiological data collected, making the system more comprehensive.

Furthermore, the authors have integrated sophisticated algorithms for the estimation of physiological parameters and applied machine learning techniques (Fuzzy Logic) for the classification of stress levels, demonstrating a leap towards intelligent health monitoring. The proposed advancements also include enhancements to both the intelligent stress assessment system and the developed mobile application. These innovations are designed to greatly assist patients in their daily lives, emphasizing practical application and user experience. Overall, the study not only introduces novel monitoring parameters but also leverages advanced analytical techniques to provide a more robust and user-friendly system for health monitoring.”

“The " 3.1. Proposed Healthcare System" section on page 4 should adequately provide a description of the overall system.”

We appreciate your comment. The reviewer has highlighted a relevant issue, namely that the Subsection 3.1. is significantly condensed. However, more detailed information can be found in the following Subsections, which are part of Subsection 3.1, namely 3.1.1, 3.1.2 and 3.1.3. Despite this, we identified a gap in the detailed information about the two monitors used, namely the Glucose Monitor and the Blood Pressure Monitor. A new Subsection “3.1.4. Glucose and Blood Pressure Monitors” has therefore been introduced, in the hope that this will address the issue raised by the reviewer.

“Further clarification should be provided under Formula 1 on page 7, and under Table 2.”

We appreciate the comment. More information was added in order to clarify the issue raised by the reviewer, and the following was added:

"Note that in Fuzzy Logic, the final process of Defuzzification requires the application of the Maximum Method or the Centroid Method, the latter being used. The variables R1, R2 and R3 refer to the indices obtained by applying the rules shown in Table 2, where each parameter of these rules refers to the Membership Degree obtained through the Membership Functions identified previously and detailed in [4]."

“In the section " 4.1. Stress Level Classification Model Update", you refer to "satisfactory results". Objects of comparison should be provided and discussed adequately.”

We appreciate the reviewer's comment. The statement has to do with the comparison between the results obtained with this new update of the proposed system compared to the previous one and presented in the work referenced as [4], conducted by the same authors. However, in order to clarify this comparison, more information on this topic has been added in Subsection 4.1, right after Table 5, in the hope of having solved the problem identified by the reviewer. For this purpose, the following has been added:

“Table 5 shows that updating the proposed fuzzy logic methodology produces satisfactory results in terms of precision, accuracy, sensitivity, and specificity, compared to the work carried out in [4], thus demonstrating that reducing the number of stress level classifications has improved the data, thus reducing dispersion, without this having influenced the system's performance.  Another motivating factor is comparing the accuracy of the proposed method with others proposed in the more detailed literature review presented in [4].”

“In section 3.1.2, the paper mentions "fast and secure authentication methods using ML technology" and also mentions "a new approach to artificial intelligence" in the title. Please fully provide and discuss the application of the system in ML.“

We greatly appreciate your comment. Subsection 3.1.2 states that a key benefit of utilising non-relational databases, like Google Firebase, is the ability to achieve exceptionally rapid access, particularly in the context of mobile or web apps. Nowhere does it indicate that ML is employed as a rapid and reliable authentication technique. In order to illustrate this, we have included the sentence in question below, with the expectation that this brief discussion will have enlightened the reviewer:

“The Google "Firebase" platform was selected for implementing this system because to its dynamic database capabilities, which offer significant benefits for mobile and web application development. It is compatible with IOS, Android, Web, Unity, and C++. This platform offers both real-time access and a high level of interaction with cloud storage, use of ML techniques, rapid and secure authentication methods, and other features [30].”

Regarding the "new approach to artificial intelligence" implied in the title of the manuscript, the work implements a Fuzzy Logic algorithm for the detection and classification of stress levels. Fuzzy Logic is a method within the field of machine learning and shares similarities with Expert Systems. In this way, the new approach using artificial intelligence is the implementation of an innovative ML (Fuzzy Logic) algorithm applied to the context of stress.

“The picture in the text is fuzzy, such as Figure 4, please check it carefully.“

We appreciate your comment. The reviewer's observation regarding Figure 4 is accurate, as it is centrally aligned inside the text. However, according to the template given to the authors, images and tables should be left-aligned. Figure 4 has been repositioned accordingly.

“There are many typographical errors. For example, on page 9, should Figure 4 be Figure 5? Please check the chart numbers and manuscript carefully.“

We appreciate the reviewer's comment. The reviewer is right, it has been noted that there are two figures labelled as Figure 4, and this has been corrected. As for the rest of the manuscript, it has been revised in order to eliminate any typographical errors that might arise, especially the numbering of the figures on the labels and in the text.

Reviewer 2 Report

Comments and Suggestions for Authors

The manuscript proposes a fuzzy logic algorithm that is based on the assessment of several indicators to estimate stress levels in diabetic patients, along with the development of a mobile application for users' stress level monitoring. While the research direction is cutting-edge and holds significant research significance, the overall coherence of the manuscript is lacking, and the description of the innovative points is unclear. Thus, rejection is recommended.

1.     The paper discusses predicting user stress levels through a sensing system (PPG/GSR/BT signals), a concept already presented in "IEEE Trans Instrum Meas, vol. 73, pp. 1–14, 2024". The novelty of this study lies in augmenting the existing computing platform with two additional channels for monitoring blood pressure and blood glucose levels. However, this platform is predominantly utilized for investigating the correlation among stress, blood pressure, and blood glucose, rather than focusing on predicting stress levels in diabetic individuals, as emphasized in the manuscript.

2.     The sources of the true and predicted stress classification in the second Figure 4 (labeling repeated within the text) and Table 5 are not sufficiently elucidated in the manuscript. This aspect of the description is overly concise and could potentially confuse readers.

3.     In Figure 5, are the “Samples” on the horizontal axis referring to different volunteers with diabetes? Is it comparing the blood glucose of one volunteer at a given time with the predicted stress classification of the same volunteer at the same time? If each data point comes from different volunteers, does the trend of increase or decrease hold any significance? Similar concerns arise regarding Figures 6 and 7.

Comments on the Quality of English Language

Please check the text carefully for grammatical errors and for contextual coherence.

Author Response

“The paper discusses predicting user stress levels through a sensing system (PPG/GSR/BT signals), a concept already presented in "IEEE Trans Instrum Meas, vol. 73, pp. 1–14, 2024". The novelty of this study lies in augmenting the existing computing platform with two additional channels for monitoring blood pressure and blood glucose levels. However, this platform is predominantly utilized for investigating the correlation among stress, blood pressure, and blood glucose, rather than focusing on predicting stress levels in diabetic individuals, as emphasized in the manuscript.”

We greatly appreciate the reviewer's comment. The reviewer is right in that the novelty of this study lies in extending the computational platform to support additional channels for monitoring blood pressure and blood glucose levels. In addition, the work also introduces an update to the algorithm for detecting and classifying stress levels. With regard to the particular problem raised by the reviewer, an attempt was made to highlight the novelty of the study in more detail, and as such, new information has been added in Section 1. Introduction, as follows:

“This study presents significant advancements to the previous research conducted by the same authors [4]. The previous work involved the development of a multichannel sensory system that focused on acquiring and processing Photoplethysmography (PPG) signals. The novelty of this study lies in the substantial expansion of the system's capabilities by incorporating the acquisition of GSR and body temperature, two parameters closely associated with stress. Moreover, the innovation in this study includes the addition of new channels for monitoring BP and blood glucose levels, which enhances the system's ability to cater to the needs of individuals with diabetes and hypertension. These novel channels significantly broaden the scope of physiological data collected, making the system more comprehensive.

Furthermore, the authors have integrated sophisticated algorithms for the estimation of physiological parameters and applied machine learning techniques (Fuzzy Logic) for the classification of stress levels, demonstrating a leap towards intelligent health monitoring. The proposed advancements also include enhancements to both the intelligent stress assessment system and the developed mobile application. These innovations are designed to greatly assist patients in their daily lives, emphasizing practical application and user experience. Overall, the study not only introduces novel monitoring parameters but also leverages advanced analytical techniques to provide a more robust and user-friendly system for health monitoring.”

One aspect that remains ambiguous and goes back to the problem identified by the reviewer, is the title of the manuscript, however, “A Novel AI Approach for Assessing Stress Levels in Patients with Type 2 Diabetes Mellitus Based on the Acquisition of Physiological Parameters Imbibed During Daily Life” goes back to the algorithm proposed for estimating stress, and in this work the algorithm already established previously is updated. The context of chronic diabetic patients also applies here, and as such, the correlation between stress and glucose was inevitable. Blood pressure is an additional factor.

“The sources of the true and predicted stress classification in the second Figure 4 (labeling repeated within the text) and Table 5 are not sufficiently elucidated in the manuscript. This aspect of the description is overly concise and could potentially confuse readers.”

We appreciate the reviewer's comment. The reviewer is absolutely right, and as such, more information has been added to the manuscript in order to clarify the issues raised, namely the second paragraph of Subsection 4.1. has been redrafted, resulting in the following:

“The performance of the model was evaluated using a multi-class confusion matrix, which is depicted in Figure 5. Note that the “True Classification” refers to the volunteers' classification of their stress levels through a self-assessment questionnaire, while the “Predicted Classification” refers to the stress levels estimated by the pro-posed Fuzzy Logic algorithm. This enables the identification of True Positives (TP), True Negatives (TN), False Positives (FP), and False Negatives (FN). Furthermore, several metrics were used to analyse the performance of the proposed stress assessment model, as outlined in Table 5. In this case, Sensitivity measures the proportion of TP that are correctly identified by the model; Specificity measures the proportion of TN that are correctly identified by the model; Precision measures the proportion of predicted positives that are correct; Accuracy measures the overall correctness of the model; F1 Score is the harmonic mean of Precision and Sensitivity, providing a balance between the two.”

“In Figure 5, are the “Samples” on the horizontal axis referring to different volunteers with diabetes? Is it comparing the blood glucose of one volunteer at a given time with the predicted stress classification of the same volunteer at the same time? If each data point comes from different volunteers, does the trend of increase or decrease hold any significance? Similar concerns arise regarding Figures 6 and 7.”

We appreciate your comment. The reviewer is right to raise this issue. The “Samples” in each graph represent a sample from each of the 128 volunteers, and as such, each point corresponds to each volunteer. However, in order to clarify this issue further, additional information was added when introducing each graph (figures 5, 6 and 7), respectively:

“Regarding the correlation of stress with blood glucose levels, the results obtained were promising, allowing to identify a clear relationship between the two parameters. In Figure 5 below, it is possible to observe the evolution of the stress levels (classified from 1 to 3 as stipulated in the proposed model) and the blood glucose levels, where Samples identifies each of the 128 participants, and being clear that when one increases, the other follow, and vice versa.”

“The correlation between blood glucose levels and blood pressure was also found to be promising, as the results enabled the identification of a distinct relationship be-tween the two parameters, considering both systolic and diastolic blood pressure components. In Figure 6 below, it is possible to observe the evolution of glucose levels and blood pressure, where Samples identifies each of the 128 participants, and to the extent that when one increases, the same is visible in the other parameter, and vice versa”

“In relation to the correlation of stress with blood pressure, the results obtained were promising, allowing to identify a clear relationship between the two parameters. In Figure 7 below, it is possible to observe the evolution of the stress levels (classified from 1 to 3 as stipulated in the proposed model) and the blood pressure represented here through its two components, i.e. systolic blood pressure and diastolic arterial pressure. Furthermore, Samples identifies each of the 128 participants."

Regarding to increase or decrease holding any significance, in reality it does, as the same classification table of physiological parameters is used for the volunteers, and so one volunteer may have, for example, a higher blood pressure value resulting in higher stress levels, while another volunteer, for example, may have a lower blood pressure value resulting in lower stress levels.

“Please check the text carefully for grammatical errors and for contextual coherence.”

We greatly appreciate your comment. The manuscript has been revised in order to eliminate any grammatical errors that might arise, and also regarding contextual coherence.

Reviewer 3 Report

Comments and Suggestions for Authors

The manuscript by Ribeiro et al. entitled "A Novel AI Approach for Assessing Stress Levels in Patients with Type 2 Diabetes Mellitus Based on the Acquisition of Physiological Parameters Imbibed During Daily Life" provides new insights, but needs some changes to improve it.

L 64-69.- Change Roman section numbers to Arabic numbers.

L70.- In addition to heart disease, there are other stress-related conditions that authors should mention, such as gastrointestinal ulcers, burnout syndrome, and Alzheimer's disease.

L.168.- The lancet used in these devices can cause stress. This is different with FreeStyle patch models. Comment and discuss.

In the discussion, the authors should comment on Respiratory Rate (RR) and Cross Time-Frequency (analyses) Widjaja D, Comput Math Methods Med. 2013;2013:451857. doi: 10.1155/2013/451857.

The values in Table 1 represent the physiological values of the healthy population. However, there are several variations of the same population that do not meet these criteria, such as high performance athletes who run at a heart rate of 40 or hyperreactors with GSR values greater than 50 kOhm. Comment and discuss.

L278.-Glucose fluctuations can occur in many situations(https://doi.org/10.2337/dc15-2035), including when subjects lie.

L284.-What are the control groups, with stress and without stress? What was the gold test to consider those with stress?

Fig 5. Does this graph include the glucose levels of all subjects tested? If so, what is the coefficient of variation for each point?

What are the limitations of this work?

Author Response

“L 64-69.- Change Roman section numbers to Arabic numbers.”

We appreciate the reviewer's comment. The reviewer is absolutely right, and as such we have corrected the error identified.

“L70.- In addition to heart disease, there are other stress-related conditions that authors should mention, such as gastrointestinal ulcers, burnout syndrome, and Alzheimer's disease.”

We greatly appreciate the reviewer's comment. The primary focus of this study is the correlation between stress and hypertension and diabetes. However, the reviewer correctly suggests that additional information on stress-related illnesses should be included to further highlight the importance of patient treatment in the presence of stress. Consequently, the initial paragraph of Section 2 has been revised to include more details regarding the effects of stress, and a new reference (reference 5) has been included, resulting in the renumbering of the references. This paragraph has been reworked, resulting in the following:

“Stress is a state of tension or pressure that results from everyday experiences or situations. It has a direct impact on the management of chronic diseases such as heart disease, asthma, obesity, diabetes, headaches, depression and anxiety, gastrointestinal problems, Alzheimer's disease, accelerated aging, premature death, among others [5]. Of these, the close link between hypertension and diabetes associated with stress stands out. Moreover, stress can be a significant burden on individuals' lives. There-fore, it is crucial to effectively manage stress levels.”

“L.168.- The lancet used in these devices can cause stress. This is different with FreeStyle patch models. Comment and discuss.”

We greatly appreciate your comment. A new subsection “3.1.4 Glucose and Blood Pressure Monitors” has been introduced, in which both glucose and blood pressure monitors are covered in greater detail, and as such we hope to have addressed the problem raised by the reviewer.

“In the discussion, the authors should comment on Respiratory Rate (RR) and Cross Time-Frequency (analyses) Widjaja D, Comput Math Methods Med. 2013;2013:451857. doi: 10.1155/2013/451857.”

We appreciate the reviewer's comment. The suggested article has been analysed and is very clear and recommended reading. However, this work does not prioritise the study of cardiorespiratory dynamics and their correlation with stress. In a prior publication by the authors, cited as [4] in the study, the authors investigated the correlation between cardiorespiratory dynamics and developed an initial version of the stress level assessment algorithm. This algorithm considered various cardiorespiratory characteristics. We appreciate the reviewer's proposal, but it has already been investigated in the previous research. The current study exclusively examines the correlation between stress, hypertension, and blood glucose levels.

“The values in Table 1 represent the physiological values of the healthy population. However, there are several variations of the same population that do not meet these criteria, such as high performance athletes who run at a heart rate of 40 or hyperreactors with GSR values greater than 50 kOhm. Comment and discuss.”

We greatly appreciate the reviewer's comment. The reviewer is right when highlights some conditions that can lead to a variation in the parameters classified in Table 1, however, these values apply to the general population and even to chronic patients. It is true that, for example, patients suffering from hyperreactors will have different GSR classifications, but the algorithm for estimating stress gets round this issue. By introducing multiple parameters, the algorithm adapts to these conditions, in this case excluding GSR values, hence the importance of patients' health conditions being considered and recorded in the mobile application. In the case of high-performance athletes, variations in classifications are also inevitable, however, at this stage of the study it is not feasible to consider all possible situations, and so we focus on the general population. In order to clarify this issue, we have added extra information to the manuscript, specifically in the last sentence before Table 1, presenting the following:

“The classification and type of Membership Function were determined based on Table 1, which displays the mean values that are representative of the overall healthy population, and they are equally applicable to individuals with chronic conditions. Variations in these parameters, for example for high-performance athletes, have not been considered due to the wide disparity in values. The aim of this work is to offer improvements in health care for the chronically ill, as well as stress management for the general population.”

“L278.-Glucose fluctuations can occur in many situations(https://doi.org/10.2337/dc15-2035), including when subjects lie.”

We appreciate the reviewer's comment. The reviewer is right, and the suggested article is well-founded. In reality, when diabetic subjects lie, this ends up inducing some changes in their conditions, for example, inducing stress, thus leading to an increase in blood glucose levels, and the same is proven in this study. In the specific part of the manuscript identified by the reviewer, it is stated that there are differences in the values for fasting and after-meal situations, because these are the two situations considered in healthcare management by healthcare professionals. These two conditions result in big changes in glucose levels, and they have to be considered, because, for example, 120mg/dL is totally normal for a diabetic after a meal, but if it's fasting, it's considered high. Other relationships, such as the one identified by the reviewer, where there is a possible relationship between lying and glucose levels, are not yet widely considered in the health field, and are only considered in studies, such as the one suggested by the reviewer, or the one proposed by us, among others. Therefore, we value the opportunity provided by the reviewer to address this issue, but we would rather focus on reliable clinical data, such as fasting and after meal classification.

“L284.-What are the control groups, with stress and without stress? What was the gold test to consider those with stress?”

We appreciate your comment. The reviewer is right to raise this issue, however, as mentioned in the manuscript, the system was developed in [4] and validated using pre-assessment questionnaires of the subjects' stress levels (gold test), thus making it possible to correlate with the stress levels estimated by the algorithm. In order to clarify the issue, in Subsection 2.4., 2nd paragraph, information has been added regarding the gold test, adding the following:

“The golden standard for assessing stress is self-assessment questionnaires, but new methods have been proposed using non-invasive physiological assessment methods, including Electrocardiography (ECG), HR, GSR and BP, as well as respiratory activity [21-23].”

Regarding to the control groups with stress or without stress, what was done and identified in the experimental protocol (Subsection 3.4.) was to collect the physiological parameters needed by the algorithm to estimate stress levels, while at the same time collecting data on blood pressure and blood glucose levels, allowing them to be correlated. Attention was paid to collecting the data at different times of the day, thus allowing the subjects to experience more relaxing or stressful situations, allowing them to self-assess their stress as calm, normal or stressed.

“Fig 5. Does this graph include the glucose levels of all subjects tested? If so, what is the coefficient of variation for each point?”

We greatly appreciate your comment. The reviewer's question is very pertinent and is based on something that is not very detailed in the manuscript. The ‘Samples’ in each graph represent a sample from each of the 128 volunteers, and as such, do not refer to some kind of average or weighted representation, thus no coefficient of variation is taken for each point. However, in order to clarify this issue further, additional information was added when introducing each graph (figures 5, 6 and 7), respectively:

“Regarding the correlation of stress with blood glucose levels, the results obtained were promising, allowing to identify a clear relationship between the two parameters. In Figure 5 below, it is possible to observe the evolution of the stress levels (classified from 1 to 3 as stipulated in the proposed model) and the blood glucose levels, where Samples identifies each of the 128 participants, and being clear that when one increases, the other follow, and vice versa.”

“The correlation between blood glucose levels and blood pressure was also found to be promising, as the results enabled the identification of a distinct relationship be-tween the two parameters, considering both systolic and diastolic blood pressure components. In Figure 6 below, it is possible to observe the evolution of glucose levels and blood pressure, where Samples identifies each of the 128 participants, and to the extent that when one increases, the same is visible in the other parameter, and vice versa”

“In relation to the correlation of stress with blood pressure, the results obtained were promising, allowing to identify a clear relationship between the two parameters. In Figure 7 below, it is possible to observe the evolution of the stress levels (classified from 1 to 3 as stipulated in the proposed model) and the blood pressure represented here through its two components, i.e. systolic blood pressure and diastolic arterial pressure. Furthermore, Samples identifies each of the 128 participants."

“What are the limitations of this work?”

We greatly appreciate the reviewer's comment. In terms of the limitations of this work, all the steps relating to future work and presented in the last paragraph of the Conclusion are in themselves limitations of this work, and as such should be addressed in the future. In order to clarify things for the reviewer, we have included the paragraph in the Conclusion that outlines the limitations and actions to consider:

“Regarding future work, one of the next steps is to enhance the system's robustness by perhaps incorporating new mechanisms and implementing changes. One of the goals is to replace the fuzzy logic technique with more powerful machine learning techniques. Furthermore, given that the purpose of this project is to assist individuals with diabetes in their everyday activities, it would be advantageous to prioritise the development of an intelligent and mobile interface that allows for seamless user inter-action. In the future, it would be beneficial to incorporate features for measuring glucose levels and blood pressure directly into the system, rather than relying on external devices currently available in the market.”

Round 2

Reviewer 2 Report

Comments and Suggestions for Authors

The author has effectively elaborated on the innovative aspects of this paper and has provided further explanations on some potentially confusing points, making it a complete and novel work overall. However, I still have a question regarding Figures 5, 6, and 7. If "Samples" refers to different volunteers, is it that the study demonstrates the correlation between blood glucose, blood pressure, and stress through the observation that one parameter increases with the increase of another in different volunteers?  If this is the case, the persuasiveness may still be insufficient. I suggest adding a study on the correlation of parameters for a single volunteer.

Comments on the Quality of English Language

Minor revision of English language required

Author Response

“The author has effectively elaborated on the innovative aspects of this paper and has provided further explanations on some potentially confusing points, making it a complete and novel work overall. However, I still have a question regarding Figures 5, 6, and 7. If "Samples" refers to different volunteers, is it that the study demonstrates the correlation between blood glucose, blood pressure, and stress through the observation that one parameter increases with the increase of another in different volunteers?  If this is the case, the persuasiveness may still be insufficient. I suggest adding a study on the correlation of parameters for a single volunteer.”

We greatly appreciate the reviewer's comment. We are pleased that we have addressed the reviewer's comments correctly, namely by effectively elaborating the innovative aspects of the paper and providing additional explanations on some potentially confusing points. As for the question mentioned regarding Figures 5, 6 and 7, the reviewer is right to deduce that the conclusions drawn came from the correlation of the increase in one parameter with another, and so on. The inclusion of samples from all the volunteers made it possible to identify patterns, thus proving that the same behaviour is observed for all the volunteers, thus making it possible to draw the conclusions presented, highlighting the strong correlation between blood glucose, blood pressure and stress. However, the reviewer is right to point out that persuasiveness may still be insufficient, emphasising the importance of also including a correlation study of parameters for a single volunteer, thus making it possible to observe the evolution of the parameters. To this end, additional information has been included, referring to a second stage of data collection carried out by some participants, in which physiological parameters were acquired in 4 phases of the day, over 5 consecutive days, thus allowing the effect of medication to be assessed (first and second phases were carried out after food intake and after the medication began to take effect) and the effect of food intake (third and fourth phases were carried out before and after food intake). This information has been added in Subsection 3.4 concerning the experimental protocol, as follows:

“To enhance the study, an additional phase was implemented for some participants, during which physiological parameters were collected four times each day, over a period of five consecutive days. The selection of these four time periods throughout the day was made in order to evaluate the impact of both meals and medicine intake. Consequently, during the first and second stages, all the physiological measurements were obtained following food intake and after the medication began to produce its effects, respectively. This enabled the evaluation of the effectiveness of the medication. In the third and fourth stages, we collected the physiological parameters both before and after food intake, respectively. This enabled us to evaluate the impact of food in-take on individuals with diabetes.”

In addition, Subsections 4.2., 4.3. and 4.4. have also been updated with textual information as well as information shown in Figures 7, 9 and 11.

“Minor revision of English language required”

We appreciate the reviewer's comment. The entire manuscript has been reviewed in detail to ensure proper usage of English as well as to identify and correct any typographical errors.

Reviewer 3 Report

Comments and Suggestions for Authors

My questions and comments have been answered by the authors. I think the manuscript "A Novel AI Approach for Assessing Stress Levels in Patients with Type 2 Diabetes Mellitus Based on the Acquisition of Physiological Parameters Imbibed During Daily Life" provides very important data and should be published.

Author Response

“My questions and comments have been answered by the authors. I think the manuscript "A Novel AI Approach for Assessing Stress Levels in Patients with Type 2 Diabetes Mellitus Based on the Acquisition of Physiological Parameters Imbibed During Daily Life" provides very important data and should be published.”

We greatly appreciate the reviewer's comment. We want to express our appreciation for the reviewer's verdict, as well as for the insightful comments made, which helped us in improving the manuscript's quality. We are pleased to have successfully fulfilled all the requested comments.
